



# Development of a chromatographic method to study oxidative potential of airborne particulate matter

Pourya Shahpoury[1,2], Tom Harner[1], Gerhard Lammel[2], Jake Wilson[2]

[1]Air Quality Processes Research Section, Environment and Climate Change Canada, Toronto, M3H 5T4, Canada
[2]Multiphase Chemistry Department, Max Planck Institute for Chemistry, Mainz, 55128, Germany

*Correspondence to*: Pourya Shahpoury (pourya.shahpoury@canada.ca)

**Abstract.** Oxidative potential (OP) is a measure of inhalation toxicity of airborne particulate matter (PM). The redox-active constituents of PM react with lung antioxidants (AOs) in the epithelial lining fluid (ELF), resulting in oxidation of AOs. The excessive loss of AOs leads to oxidative stress, inflammation of the epithelial tissue, and chronic diseases. In this work, we developed a novel rapid chromatographic method, employing an ultra-high performance liquid chromatograph coupled to a triple-quadruple mass spectrometer (UHPLC-MS/MS), to determine the OP of ambient PM, and investigated the application of electrochemical oxidation-reduction potential (ORP) as an alternative approach for estimating OP. We measured the direct oxidation of AOs, ascorbic acid, glutathione, and cysteine, and formation of glutathione disulfide and cystine, following PM addition to various simulated ELF (SELF) formulations which, in addition to AOs, contained inorganic salts, a phospholipid, and proteins. The assay performance was evaluated using standard reference PM, and we investigated the links between OP dose-response, time-dependence, and the ORP. The new assay showed a high precision, and when applied to PM, OP and ORP increased with both reaction time and PM concentrations in SELF. The presence of SELF inorganic species and surfactant lipid increased the OP determined through oxidation of glutathione and cysteine, but showed an opposite effect with ascorbic acid. The presence of proteins did not affect the OP. The findings suggest that ORP measurement could be used as an alternative and simple approach for estimating the OP of ambient PM.





## 1 Introduction

Air pollution is a major global health risk. It has been associated with respiratory and cardiovascular diseases as well as increased morbidity and mortality rates over the past 25 years. Human exposure to fine particulate matter (PM), specially PM with aerodynamic diameter ≤2.5 µm, is one of the main causes of the adverse effects across the world
(WHO, 2016; Cohen et al., 2017; Burnett et al., 2018; Lelieveld et al., 2018; Lelieveld et al., 2019). Recent studies suggested that exposure to fine PM significantly reduces the human life expectancy in both low-middle income and developed countries (Lelieveld et al., 2018; Lelieveld et al., 2019a), with traffic-related emission being responsible for 65% of premature death from anthropogenic air pollutants worldwide (Lelieveld et al., 2019b).

For decades, PM mass concentration has been applied as a metric to relate air pollution with adverse health outcomes;
however, a large part of PM is made of non-toxic components such as inorganic salts and crustal dust. In fact, only a small portion of PM, including the organic phase and metals, can pose toxic effects (Ayres et al., 2008; Lodovici and Bigagli, 2011). Moreover, the pool of chemicals in PM is dynamic as it undergoes temporal and spatial changes due to reaction with oxidants and mixing in the atmosphere (Paula et al., 2016). Oxidative potential (OP), as opposed to PM mass concentration, is known to be a better indicator of PM inhalation toxicity (Ayres et al., 2008). OP is defined
as the ability of chemicals to pose oxidative stress. Through this process, the redox-active constituents of PM, e.g. quinones and transition metals (TMs), react with lung antioxidants (AOs) in the epithelial lining fluid (ELF), resulting in depletion of AOs and formation of AO oxidation products. In case of thiol-containing AOs, such as glutathione (GSH) and cysteine (CSH), the oxidation leads to formation of oxidized glutathione (GSSG) and cystine (CSSC) through formation of a disulfide bridge (Figure 1). Excessive loss of lung AOs leads to oxidative stress, which follows
the formation of free radicals, e.g. $O_2^{-\cdot}$, $\cdot OH$, $RO^{\cdot}$, $ROO^{\cdot}$, and non-radical species such as $O_3$ and $H_2O_2$ in the ELF (Kelly and Fussell, 2017; Sies et al., 2017). Although quinones and TMs are widely recognized as being redox-active, other redox-active chemicals, such as those in humic-like substances, are also anticipated to pose OP (Charrier and Anastasio, 2012; Verma et al., 2015; Dou et al., 2015; Gonzalez et al., 2017).

To this date, OP has been primarily determined using colorimetric methods, and the dithiothreitol (DTT) assay has
been used most frequently (Charrier and Anastasio, 2012; Charrier et al., 2014; Verma et al., 2009; Verma et al., 2014; Verma et al., 2015; Calas et al., 2017; Crobeddu et al., 2017; Calas et al., 2018; Tong et al., 2018; Tuet et al., 2019). Such methods introduce certain challenges; for instance, DTT is a chemical that is alien to ELF and the assay was shown to have greater response to PM-bound quinones rather than metals (Ayres et al., 2008; Charrier et al., 2014); metals outweigh quinones in ambient PM. It was suggested that the DTT oxidation rates may not correspond entirely
to the formation rates of reactive oxygen species (Xiong et al., 2017). Moreover, DTT oxidation demonstrated degrees of association to changes in PM components, sources (e.g. biomass burning, traffic), and temporal variations that were different from those seen with GSH or ascorbic acid (AA) (Fang et al., 2016; Weichenthal et al., 2019). Such observations raise the question of whether DDT oxidation rate is a realistic metric for OP of ambient PM in physiological ELF.

Colorimetric methods are not fully compatible with ELF containing lipids and proteins because the turbidity resulting from their presence in ELF could interfere with reading of UV probe. Extraction of PM in simulated-ELF (SELF), as opposed to water, as well as SELF formulation could in various ways affect the OP results (Calas et al., 2017). The



presence of lipids and proteins could in particular be important as they influence the ELF characteristics: for instance, dipalmitoylphosphatidylcholine (DPPC), a major ELF surfactant phospholipid, is known to reduce the ELF surface tension (Gregory et al., 1991; Griese, 1999; Johansson et al., 1994; Pison et al., 1986; Boisa et al., 2014). The presence of DPPC was previously linked to improved OP dose-response relationships, supposedly by enhancing suspension

stability (Calas et al., 2017). Albumin, a major ELF protein (Bredberg et al., 2012), could bind to metals and limit their solubility; hence, influencing ELF antioxidant and PM pro-oxidant capacity (Cross et al., 1994).

Chromatographic methods have been applied less frequently to studies of OP from air samples (Mudway and Kelly, 1998; Zielinski et al., 1999; Künzli et al., 2006; Crobeddu et al., 2017; Calas et al., 2018). These studies determined the consumption of AA in a simplified SELF (0.9% saline, pH 7.4, no apparent buffering capacity) using high

performance liquid chromatograph coupled with electrochemical detection (Iriyama et al., 1984). Crobeddu et al., (2017) advanced this method by measuring GSH and GSSG in addition to AA; however, they did not take into account autoxidation of GSH during sample processing. Autoxidation was shown to be a major source of uncertainty in measuring OP using thiol-containing AOs in biological samples (Giustarini et al., 2016). The potential autoxidation could lead to low reproducibility of measured GSH and GSSG concentrations, as was previously seen (Crobeddu et

al., 2017).

The existing measures of OP face two key disadvantages. Firstly, it is a time-consuming and expensive task to generate OP measurements for the wide range of chemicals present in PM. Secondly, for the same chemical species, different assays may give varying responses (Bates et al., 2019). In an attempt to reconcile these two limitations, the use of the electrochemical oxidation-reduction potential (ORP) may provide an alternative or complimentary approach. The

standard reduction potential is a physicochemical property that quantifies the ability of a chemical to transfer electrons (von Smolinski et al., 1989; Rajeshwar and Ibanez, 1997). Given that AOs and surrogate molecules used in assays rely on reactions involving the transfer of a single electron, one would expect the reduction potential to be a useful predictor of OP. Standard reduction potential data is available for a wide range of organic compounds (Vanýsek, 2002). This data is used in many fields: from battery research (Palacín, 2009) to enology (Danilewicz et al., 2019). In

addition, quantum mechanical methods allow reduction potentials in water to be calculated theoretically; such techniques have already been used on atmospherically relevant compounds such as quinones (Lynch et al., 2012; Guerard and Arey, 2013; Marenich et al., 2014; Méndez-Hernández et al., 2015). In the biochemical literature, the Nernst equation has been frequently used to calculate the reduction potential of antioxidant redox couples such as GSH/GSSG (Schafer and Buettner, 2001). Despite some controversy about the interpretation of these reduction

potentials (Flohé, 2013), such values have been used as measure of the oxidation environment within cells.

In the present work, a new and rapid chromatographic method was developed to determine the OP using an ultra-high performance liquid chromatograph (UHPLC) coupled to a triple-quadruple mass spectrometer (MS/MS). This method measures the direct consumption of AA, CSH and GSH and formation of CSSC and GSSG following incubation of PM in SELF which, in addition to AOs, contains major ELF inorganic salts, a phospholipid, and proteins. We took

specific measures to prevent autoxidation of thiol-AOs. The method performance and reproducibility, the OP time-dependence and dose-response, and the effect of SELF formulation on OP were investigated. In addition, we examined



the relationship between OP and ORP, an electrochemical measure of the assay mixture, and compared this to a theoretical value based on the Nernst equation and antioxidant concentrations.

## 2 Methods

### 2.1 Chemicals

AA, GSH, GSSG, and isotopically labelled AA-$^{13}$C$_6$, GSH-$^{13}$C$_2$$^{15}$N, CSH-$^{13}$C$_3$, and GSSG-$^{13}$C$_4$$^{15}$N$_2$ were purchased from Toronto Research Chemicals (Toronto, Canada). CSH, CSSC, N-ethylmaleimide (NEM), DPPC, ethylenediaminetetraacetic acid (EDTA), sulfosalicylic acid (SSA), nitric acid, phosphate buffered saline (PBS; Bio-Performance, containing NaCl, KCl, and Na$_2$HPO$_4$), calcium chloride (CaCl$_2$), magnesium chloride (MgCl$_2$), sodium sulphate (Na$_2$SO$_4$), glycine, uric acid (UA), and albumin were purchased from Sigma Aldrich (Oakville, Canada).

CSSC-d$_6$ was purchased from TLC Pharmaceutical Standards (Aurora, Canada). LC-MS grade acetonitrile and methanol were obtained from Caledon Laboratory Chemicals (Georgetown, Canada), and LC-MS grade (Optima) water and formic acid were purchased from Fisher Scientific (Ottawa, Canada). The stock solution of analytical standards AA, AA-$^{13}$C$_6$, GSH, GSH-$^{13}$C$_2$$^{15}$N, CSH, CSH-$^{13}$C$_3$, GSSG, GSSG-$^{13}$C$_4$$^{15}$N$_2$ were prepared in 50:50 methanol-H$_2$O mixture, whereas CSSC and CSSC-d$_6$ were prepared in 10:90 methanol-H$_2$O; these were used only for

preparing calibration standards. In addition, a 50 mM mixture of AA, GSH, and CSH was prepared in 20:80 methanol-H$_2$O and used specifically to prepare SELF. All AO stock solutions were stored at -80°C. The 20% methanol solution allows swift thawing (~1 min) of concentrated AO mixture at room temperature and helps prevent autoxidation of AOs due to long thawing times prior to the experiments. 100 mM stock solution of NEM and a solution containing EDTA (2 mM) and SSA (2%) were prepared in 20:80 methanol-H$_2$O and stored at 4°C. We avoided glass-ware for

sample processing and all plastic-ware used in this study were cleaned with Nalgene L900 soap, 0.1 M nitric acid, deionized water, and LC-MS grade methanol.

### 2.2 Sample characteristics

#### 2.2.1 Particulate matter

Standard reference material (SRM 1649b) was obtained from the National Institute of Standard and Technology

(NIST, Gaithersburg, USA). It represents the typical urban PM. The sample has been characterized for a large number of organic and inorganic species, including some of redox-active substances, i.e. 9,10-anthraquinone (1.8 pg μg$^{-1}$), 1,2-benzanthraquinone (3.6 pg μg$^{-1}$), copper (311 pg μg$^{-1}$), and manganese (337 pg μg$^{-1}$). In terms of mass size distribution, the volume weighed mean falls in the range of coarse atmospheric particles. In the present study, SRM suspension in LC-MS grade water was freshly made (5 mg mL$^{-1}$) and used across different experiments. The PM

suspension was gently mixed each time prior to spiking into SELF.





### 2.2.2 SELF formulations

Different SELF formulations representing extracellular fluid of human lung were tested in the present study; a simplified formula was used as the basis for all formulations and contained PBS (pH: 7.4) and a mixture of CSH, GSH, and AA (200 µM each). These salts and antioxidants are among constituents of Gamble solution and physiological ELF (Stopford et al., 2003; Boisa et al., 2014). Four other formulations were used in order to explore the effects of SELF composition on OP results. These formulas, in addition to the above components, contained UA, $CaCl_2$, $MgCl_2$, $Na_2SO_4$, DPPC, albumin, and glycine at physiological concentrations (Boisa et al., 2014). The details of SELF formulations are indicated in Table 1. The presence of inorganic salts, antioxidants, surfactant lipids, and proteins may affect the dissolution kinetics of redox-active species (Boisa et al., 2014). The pH of SELF formulations was monitored during incubation in an independent experiment.

### 2.3 Sample preparation

The stock solution of reduced AOs (50 mM) was swiftly thawed (~1 min) at room temperature before each experiment and spiked into SELF to achieve AO concentration of 200 µM each using positive displacement pipettes (Microman E, Gilson, Middleton, USA). 2.5 mL of SELF was transferred to pre-cleaned 8-mL low-density polyethylene (LDPE) bottles (Nalgene, Thermo Scientific, Waltham, USA). LDPE bottles containing SELF were spiked with PM stock suspension in order to achieve PM concentrations ranging from 10 to 80 µg mL$^{-1}$. The samples containing PM were gently mixed and incubated along with reference SELF that did not contain PM. The incubation was performed at 37°C and 60 rpm using an incubator-shaker (Benchmark Scientific, Sayreville, USA) at intervals ranging from 30 to 240 min. Following the incubation, 300-µL aliquots ($n = 3$) of each sample were transferred to 1.5-mL centrifuge tubes (Brand, Wertheim, Germany), added with 100 µL of 100 mM NEM, mixed for 5 sec using a vortex mixer, and allowed to react at room temperature for 1 min. This reaction time is sufficient for derivatization of thiols (Escobar et al., 2016). The sample was subsequently added with 200 µL of a mixture containing 2% SSA and 2 mM EDTA (Moore et al., 2013), mixed for 10 secs, and centrifuged (10000 g, 6 min). Finally, 5 µL of the supernatant from each sample was transferred to a push-filter vial with a polytetrafluoroethylene filter (Whatman, Pittsburgh, USA) containing 485 µL of 20:80 methanol-H$_2$O and spiked with 10 µL of labelled AO standard mixture (10 ng µl$^{-1}$). The SSA in the precipitation mixture was used to precipitate albumin and to reduce the pH to ~2, whereas EDTA was used to chelate transition metals (Moore et al., 2013). We applied this mixture across all SELF formulations in order to ensure consistency in sample processing. The ORP was measured in SELF-a (containing PBS and AOs only; see Table 1) for experiments related to the assay time-dependence and dose-response. This was performed using a platinum ORP electrode with ARGENTHAL reference system (InLab Redox Micro, Mettler Toledo, Giessen, Germany). The electrode performance was verified prior to each experiment using a redox buffer (220 mV, pH 7; Mettler Toledo).

### 2.4 Sample and data analysis

The sample analysis was carried out using an Acquity UHPLC (Waters, Milford, USA) coupled to a Xevo TQ-S MS/MS (Waters). The analysis was performed in electrospray ionisation in the negative mode for AA and AA-$^{13}C_6$





and in the positive mode for the rest of the analytes. The auto-sampler temperature was set to 4°C and the injection volume was 2 µl. The target substances were separated using an Acquity HSS T3 UPLC column (3 mm, 50 mm, 1.8 µm; Waters), connected to a pre-column (2.1 mm, 5 mm, 1.8 µm, Waters), and thermostated at 40°C. The mobile phase A was composed of LC-MS grade $H_2O$ containing 0.1% formic acid, and mobile phase B was made of LC-MS

grade acetonitrile containing 0.1% formic acid. The mobile phase flowrate was set to 0.4 mL min[-1]. For the target compounds in the negative mode, the mobile phase gradient started at 20% phase B, ramped to 100% B from 0.3 to 0.5 min and held for 4.5 min, then decreased to 20% B from 5 to 5.1 min and held for 2.9 min. For substances in the positive mode, the gradient started at 1% phase B, ramped to 50% B from 0.1 to 0.3 min and held for 3.2 min, followed by an increase to 100% B over 0.1 min and a hold time of 2.4 min, followed by a decrease to 1% B and re-equilibration

time of 1.9 min.

The target compound detection parameters are presented in Table 2. The following MS/MS parameters were used: capillary voltage of 1.5 kV, source temperature of 120°C, nitrogen and desolvation gas flow-rates of 150 and 900 L hr[-1], desolvation temperature of 500°C, and collision gas flow rate of 0.15 mL min[-1]. The analyte quantification was done using the internal calibration method with 6-point linear calibration curves ($r^2 = 0.999$) ranging from 1 to 250 pg

µl[-1]. The chromatographic data analysis was performed using MassLynx software (Waters). Blanks were analysed after the samples and highest-level calibration standard, and analyte carry-over was below the instrument detection limits ($n = 3$).

AO consumption (%) was calculated as follows: $(CSH_{REF} - CSH_{PM})/CSH_{REF}\times100$, where $CSH_{REF}$ and $CSH_{PM}$ are the measured concentrations (µM) of CSH in the reference SELF and SELF containing PM after the incubation,

respectively. The consumption of AA and GSH were also calculated using the above formula. The oxidation product formation (%) was calculated as follows: $(CSSC_{PM} - CSSC_{REF})/(CSH_{REF}/2)\times100$, where $CSSC_{PM}$ and $CSSC_{REF}$ denote the measured concentrations of CSSC in SELF containing PM and the reference SELF after the incubation, respectively. Note that the second formula calculates % formation of CSSC (due to the effect of PM) relative to the total amount of CSSC that would be produced if the entire CSH in SELF were converted to CSSC following the

reaction stoichiometry of 2 CSH → 1 CSSC. The formation of GSSG was calculated using the same formula.

The theoretical redox state of CSH/CSSC and GSH/GSSG redox pairs, Eh (mV), were determined using the measured molar concentrations (mol L[-1]) of these species following the Nernst equation, Eq. 1 and 2 (Jones et al., 2000; Iyer et al., 2009):

$$Eh_{(CSH/CSSC)} = -250 + 30\times\log([CSSC]/[CSH]^2) \qquad\qquad\text{Eq. 1}$$

$$Eh_{(GSH/GSSG)} = -265 + 30\times\log([GSSG]/[GSH]^2) \qquad\qquad\text{Eq. 2}$$

## 3 Results

### 3.1 Method performance

The method reproducibility was tested using SELF-a (see Table 1 for details). Four samples containing PM (20 µg mL[-1]) were incubated together with a reference sample for the duration of 180 min. Three replicates were prepared



from each SELF sample containing PM and five replicates prepared from the reference SELF. The mean consumption for AA, CSH, and GSH were 33±3, 45±4, and 50±4%, respectively, with relative standard deviation (%RSD) ≤9% (Figure 2A). These values reflect the *inter-sample* reproducibility, targeting the entire sample preparation procedure. The *intra-sample* variabilities ($n$ = 3) were smaller, with %RSD ≤3%. The formation of CSSC and GSSG were

similarly reproducible, i.e. 11±1 and 17±1%, with inter- and intra-sample RSD of ≤13 and ≤6%, respectively (Figure 2B). The results indicate noticeable enhancement in precision compared to those reported by Crobeddu et al., (2017), where autoxidation of thiol-AOs was not considered. High reproducibility is essential for statistically valid comparison of OP between different samples. Our results further confirm the need for derivatization when analysing thiol-AOs (Giustarini et al., 2016). We investigated if our post-derivatization sample processing influenced the OP results: in a

separate experiment, which was conducted with PM in SELF-a, we replaced the precipitation solution (i.e. 2 mM EDTA/2% SSA) with (a) $H_2O$ and (b) 2 mM EDTA (note that treatment with EDTA/SSA is necessary for effective deposition of proteins prior to instrumental analysis (Moore et al., 2013), when these are present in SELF formulation). The OP was found to be independent of the processing methods mainly due to the use of internal standards in our method, which accounted for post-incubation changes in concentrations of target compounds in both reference SELF

and SELF containing PM. Among the three treatments, %RSD of the AO depletion and oxidation product formation were ≤2 and ≤5%, respectively. The addition of EDTA/SSA mixture, however, prevented the *absolute* loss of AA by ≤17% in both reference SELF and SELF containing PM, compared to the other two sample treatments. This indicates that the addition of EDTA/SSA helps stabilize AA. There was no noticeable difference in the *absolute* abundance of the other target analytes, which is reasonable given that CSH and GSH underwent derivatization immediately

following the incubation. Treatment with EDTA/SSA resulted in slightly higher reproducibility, particularly in the formation of CSSC and GSSG, i.e. ≤3% smaller %RSD.

### 3.2 Assay kinetics

The assay reaction kinetics were studied using SELF-a at PM concentration of 25 µg mL$^{-1}$ (Figure 3). The consumptions of AOs were similar in the first 60 min (11±1%; $n$ = 3); however, beyond this time point, CSH and

GSH diverged from AA at increasing rates, and this trend continued with passing of time (Figure 3A). CSH and GSH showed similar consumptions that were constantly higher than that for AA with a difference ranging from 2 to 17%. The highest AO consumptions were found at 240 min interval and were 69.2±0.5% for CSH, 69.1±0.4 for GSH and 51.9±0.5% for AA (Figure 3A).

Figure 3B shows the % formation of CSSC and GSSG. These values were similar at 30 min time-step, i.e. ≤2%, after

which they grew apart at increasing rates. The highest formations were seen at 240 min interval and were 17.2±0.3% and 26.8±1.3%, respectively (Figure 3B). Considering the molar concentrations of the redox pairs CSSC/CSH and GSSG/GSH, the observed formations of CSSC and GSSG were found to be noticeably lower than the theoretical values anticipated from reaction stoichiometry, i.e. 2 CSH → 1 CSSC and 2 GSH → 1 GSSG. This may be related to the formation of complexes between the organic molecules in PM and deprotonated GSH or CSH, prior to formation

of CSSC or GSSG, such as glutathionylated quinone species (Song and Buettner, 2010).



Figure 3C shows the ORP measurements in reference SELF ($ORP_{REF}$) and SELF containing PM ($ORP_{PM}$) at each time-step, as well as $ORP_\Delta$ ($ORP_{PM} - ORP_{REF}$). $ORP_{REF}$ values showed a decreasing trend, indicating a reducing condition throughout the incubation. The $ORP_{PM}$ values, on the other hand, were constantly higher than $ORP_{REF}$, indicating the prevailing presence of oxidants in PM. $ORP_{PM}$ and $ORP_{REF}$ showed a diverging trend throughout the

incubation, particularly at 120 min time-step when $ORP_{PM}$ started to increase – a trend completely opposite to that seen for $ORP_{REF}$. The largest gap was seen at 240 min time point, with $ORP_{REF}$ of $-56.5\pm0.4$ and $ORP_{PM}$ of $-22.2\pm0.4$ mV. As can be seen in Figure 3A-C, $ORP_\Delta$ increased exponentially with the reaction time, indicating that the SELF redox state became more oxidising following the redox reactions that led to increased depletion of AOs.

### 3.3 Effect of PM concentration

The response characteristics of the assay to various doses of PM were studied using SELF-a (see Table 1), PM concentration ranging from 10 to 80 µg mL$^{-1}$, and incubation time of 180 min. As shown in Figure 4A-B, the OP showed an increasing trend with PM concentration. GSH demonstrated the highest consumption, followed closely by CSH, and with a relatively larger gap by AA. This pattern was evident at PM concentration of $\leq$20 µg mL$^{-1}$; however, beyond this range, the depletions of CSH and GSH were found to be nearly the same ($\geq$91%) and noticeably higher

than that of AA (62-91%; Figure 4A). The highest increase in OP was between PM concentration of 20 and 40 µg mL$^{-1}$, and this was relatively more pronounced for CSH and GSH (from 49 to 92%) than AA (from 36 to 62%). The CSH and GSH depletions were near-linear in the PM concentration of 10-40 µg mL$^{-1}$, slowed down beyond this point, and levelled off at $\geq$ 60 µg mL$^{-1}$ (Figure 4A). Similar response of CSH and GSH to PM suggests that CSH could be alternatively used to address the OP of ambient PM; this has not been considered in the past.

The AA depletion covered a larger linear range, i.e. up to PM concentration of 60 µg mL$^{-1}$, but it slowed down beyond this concentration, with the highest depletion (91%) found at 80 µg mL$^{-1}$ (Figure 4A). Although the linear response range of AOs could depend on PM composition and mass-size distribution, the current results are useful when normalizing the OP to PM mass. The linearity in its response suggests that AA might be a better candidate when studying the ambient PM that varies in concentration widely across different types of sites. AA was also shown to

respond to both Fe and Cu, along with quinones, whereas GSH had a negligible response to Fe (Ayres et al., 2008). Similar to AOs, the % formation of CSSC and GSSG increased with PM concentration (Figure 4B) from 10 to 60 µg mL$^{-1}$ and plateaued beyond this range. The observed formations were relatively smaller from 40 to 60 µg mL$^{-1}$, in agreement with depletion values seen for CSH and GSH. It is interesting to note that the formations of CSSC and GSSG were smaller than that anticipated from reaction stoichiometry, as we also found with the results of the kinetic

study (Section 3.2). As shown in Figure 4A-B, the measured $ORP_\Delta$ values also increased with PM concentration and, consequently, depletion of AOs and formation of oxidation products. The increase in $ORP_\Delta$ was moderate (9 mV) in PM concentration range of 10-20 µg mL$^{-1}$, it showed a sharp increase (78 mV) at 20-60 µg mL$^{-1}$, and relatively smaller increase (17 mV) beyond this concentration range.

We estimated the theoretical redox state of CSH/CSSC and GSH/GSSG redox pairs (Eh) using the Nernst equation

(Eq. 1 and 2 in Section 2.4). As shown in Figure 5, the values of $ORP_\Delta$ and $Eh_\Delta$ agree with one another in that they both increase with PM concentration and OP, although the measured $Eh_\Delta$ is somewhat higher than $ORP_\Delta$. This





relationship is linear for CSH/CSSC redox-pair and near-linear for GSH/GSSG pair up to the PM concentration of 60 µg mL$^{-1}$. Beyond this concentration, Eh$_\Delta$ levels off due to complete consumption of CSH and GSH over the reaction time (Figure 5). This observation is particularly interesting because (1) it shows that OP of PM can be presented as Eh, a more meaningful parameter that considers a redox pair as opposed to a single antioxidant, and (2) it suggests that measuring the ORP of SELF could be an alternative and simple method for studying the OP. More samples with varying chemical composition, spatial and temporal distribution need to be studied in order to verify this possibility; this is one aspect of our ongoing work.

### 3.4 SELF formulations

The effect of SELF composition on OP was examined using SELF formulations that are listed in Table 1. As shown in Figure 6A-B, the addition of UA (SELF-b) did not noticeably affect the depletion of AOs or formation of CSSC; however, the mean formation of GSSG was found to be 6% higher in SELF-b than SELF-a. The presence of additional inorganic salts, CaCl$_2$, MgCl$_2$, and Na$_2$SO$_4$ (SELF-c) led to increase in depletion of CSH and GSH by ≤13% and formation of CSSC and GSSG by ≤6% (Figure 6A-B). This increase may be related to enhanced electron circulation in the redox system due to the presence of additional electrolytes, or due to an increase in the SELF ionic strength, enhancing the solubility of PM-bound transition metals. Interestingly, AA showed an inverse response to this change in SELF formulation, and its depletion was 11% smaller in SELF-c than in SELF-b. Further changes in AA depletion due to variation in SELF composition were not observed (Figure 6A). In contrast, the addition of ELF surfactant lipid DPPC (SELF-d) resulted in a slight increase in depletion of CSH and GSH (≤7%) as well as formation of CSSC and GSSG (≤5%). We did not observe noticeable changes in OP with the addition of ELF proteins albumin and glycine (SELF-e). In order to ensure the stability of pH in the SELF mixtures, in a separate experiment, we measured the pH in all reference SELF mixtures and SELF mixtures containing PM (Table 1) over 180 min reaction time. The pH was found to be 7.3±0.1; hence, no noticeable effect of pH on the OP results can be anticipated. Overall, our results show that, with the exception of AA, the OP that is measured through depletion of AOs can be ≤20% higher in the presence of a SELF which, in addition to AOs, contains major inorganic salts, lipid, and proteins. This suggests that, in order to determine a more realistic OP, a SELF that better represents the true human ELF should be used, rather than the simplified formulas that have been used in the past.

### 4 Conclusions

In summary, we developed a method that uses UHPLC-MS/MS to measure the OP of ambient PM. We included CSSC/CSH redox pair, measured ORP in SELF, and included physiologically relevant ELF components, i.e. inorganic salts, a phospholipid, and proteins. This was never done before to address the OP of air samples. The developed method showed high reproducibility of the measured OP, both inter- and intra-sample, when compared to previous methods, highlighting the importance of derivatization when analysing thiol-containing AOs, as well as the use of internal standards to account for variations during instrumental analysis. The oxidation of AOs, formation of oxidation products, and the ORP of SELF increased with the reaction time and concentration of PM. The similarity of response



between CSH and GSH suggests that the CSH oxidation rates can be used as an alternative to determine OP. The advantage of CSH is that its labelled standard is more readily available compared to GSH. Given the initial AO concentration of 200 μM, the AO responses were linear up to PM concentration of 60 μg mL$^{-1}$, with AA covering a relatively wider concentration range. Although the linear range might be different in the presence of different

quantities of the AOs and PM redox-active substances, the current observation suggests that AA is a better candidate when normalizing OP to PM mass, particularly when working with PM samples that vary widely in concentration. Our results showed that the presence of additional electrolytes and DPPC could influence the measured OP in various ways, increase the oxidation of thiol-containing AOs and decrease that for AA, whereas proteins may not have noticeable effects. This work suggests that ORP measurement could be used as an alternative and simple approach for

estimating the OP of ambient PM. Further work needs to be done to fully realize the potential of ORP measurement for estimating OP, by including samples that represent the chemical composition of various types of atmospheric aerosols and their spatial and temporal variations.

*Acknowledgements*

This work was funded by Chemical Management Plan, Oil Sands Monitoring, Health Canada (ATOUSSA project), and Climate Change Action Plan. We thank Ky Su, Anita Eng, and Cassie Rauert for their support with the laboratory work.

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





**Table 1.** Details of SELF formulations

|  | SELF-a | SELF-b | SELF-c | SELF-d | SELF-e | mg L$^{-1}$ |
|---|---|---|---|---|---|---|
| NaCl[a] | * | * | * | * | * | 8065 |
| KCl[a] | * | * | * | * | * | 201 |
| Na$_2$HPO$_4$[a] | * | * | * | * | * | 1420 |
| CaCl$_2$[b] |  |  | * | * | * | 256 |
| MgCl$_2$[b] |  |  | * | * | * | 200 |
| Na$_2$SO$_4$[b] |  |  | * | * | * | 72 |
| AA[c] | * | * | * | * | * | 35 |
| CSH[c] | * | * | * | * | * | 24 |
| GSH[c] | * | * | * | * | * | 62 |
| UA[b] |  | * | * | * | * | 16 |
| DPPC[b] |  |  |  | * | * | 100 |
| Albumin[b] |  |  |  |  | * | 260 |
| Glycine[b] |  |  |  |  | * | 376 |

[a] composition of standard PBS used (pH 7.4); [b] follows reported physiological concentrations (Boisa et al., 2014); [c] 200 µM of each AO was used

**Table 2.** LC-MS/MS detection parameters for target substances

| Analytes | Parent ion $m/z$ | Daughter ion $m/z$ | Collision energy | Cone voltage | Retention time (min) |
|---|---|---|---|---|---|
| AA | 175 | 115 | 12 | 6 | 0.62 |
| AA-$^{13}$C$_6$ | 181 | 119 | 12 | 6 | 0.62 |
| CSH-NEM | 247 | 158 | 22 | 14 | 1.61 |
| CSH-$^{13}$C$_3$-NEM | 250 | 158 | 20 | 25 | 1.61 |
| CSSC | 241 | 120 | 18 | 2 | 0.62 |
| CSSC-d$_6$ | 247 | 155 | 14 | 18 | 0.61 |
| GSH-NEM | 433 | 201 | 20 | 10 | 1.61 |
| GSH-$^{13}$C$_2$$^{15}$N-NEM | 436 | 201 | 22 | 26 | 1.61 |
| GSSG | 613 | 355 | 22 | 18 | 1.60 |
| GSSG-$^{13}$C$_4$$^{15}$N$_2$ | 619 | 361 | 22 | 18 | 1.60 |





**Figure 1.** Oxidation of CSH through reaction with quinones (Q) and transition metals ($Fe^{3+}$, $Cu^{2+}$) and subsequent formation of CSSC, semi-quinones ($SQ^{-}$) and reduced transition metals ($Fe^{2+}$, $Cu^{+}$).





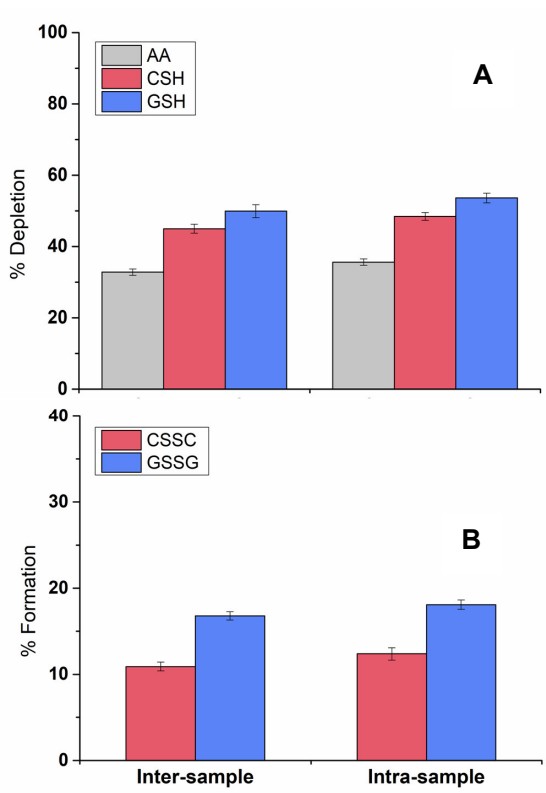

**Figure 2.** (A) percent depletion of antioxidants, AA, CSH, and GSH, and (B) percent formation of oxidation products CSSC and GSSG in SELF-a. *Inter*-sample reflects the reproducibility of entire method ($n = 4$; PM concentration of 20 µg mL$^{-1}$), whereas *intra*-sample shows the post-incubation variability due to sample processing (from replicates of a single sample; $n = 3$; PM concentration of 20 µg mL$^{-1}$).

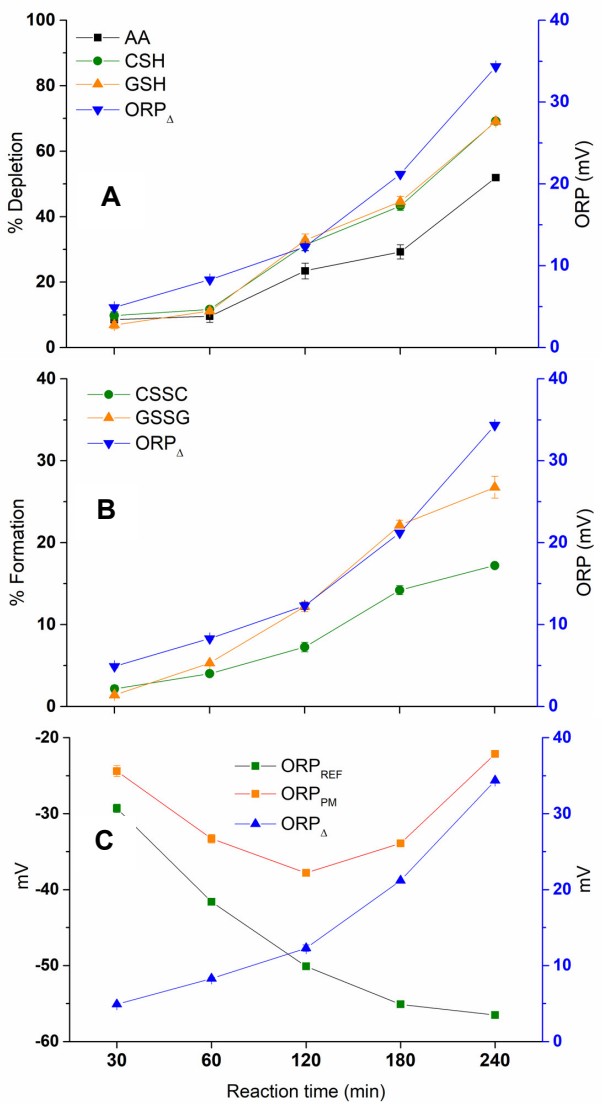

**Figure 3.** Time-dependence of antioxidant depletion (A; left y-axis), oxidation product formation (B; left y-axis), and corresponding changes of $ORP_\Delta$ (mV; right y-axis in A-C), where $\Delta$ denotes the difference in ORP values between SELF containing PM ($ORP_{PM}$; left y-axis in C) and reference SELF ($ORP_{REF}$).

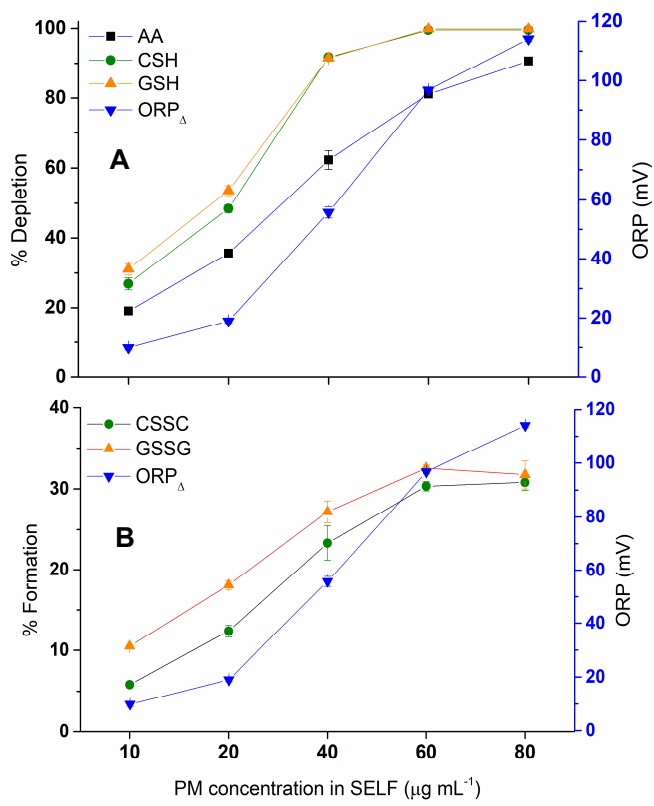

**Figure 4.** Dose-response due to increase in PM concentration in SELF (A and B, x-axis); antioxidant depletion (A; left y-axis), oxidation product formation (B; left y-axis), and corresponding changes in $ORP_\Delta$ (mV; right y-axis in A-B), where $\Delta$ denotes the difference in ORP between SELF containing PM and reference SELF.





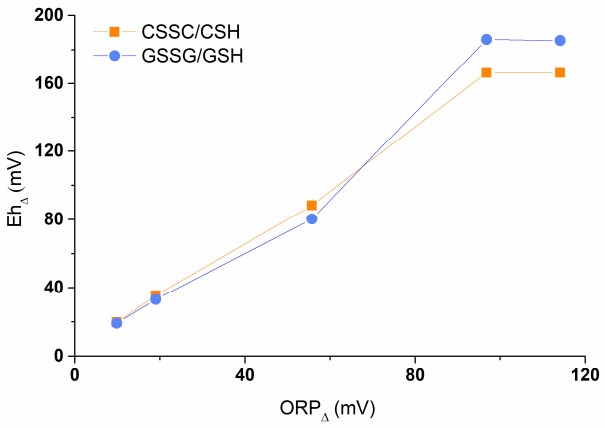

**Figure 5.** The relationship between $Eh_\Delta$ (theoretical redox state for CSSC/CSH and GSSG/GSH redox pairs) and $ORP_\Delta$, where $\Delta$ denotes the difference in Eh or ORP values between SELF containing PM and reference SELF. Note that $Eh_\Delta$ and corresponding $ORP_\Delta$ values increase with PM concentration in SELF.





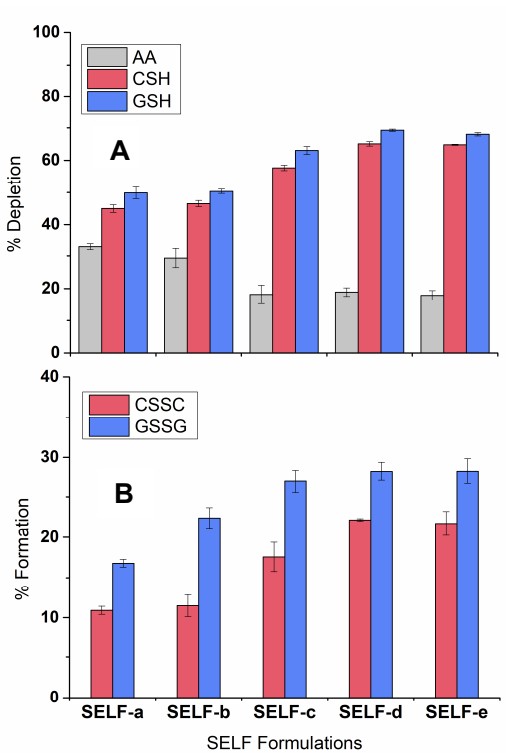

**Figure 6.** The effects of SELF formulation on antioxidant depletion (A) and oxidation product formation (B). The details of SELF formulations are given in Table 1.