# Peer review of "Development of an antioxidant assay to study oxidative potential of airborne particulate matter"

_Atmospheric Measurement Techniques, 2019_

## Referee Comment (RC1) · Anonymous Referee #3 · 1 Jul 2019

The comment was uploaded in the form of a supplement:
https://www.atmos-meas-tech-discuss.net/amt-2019-220/amt-2019-220-RC1-supplement.pdf

---

## Referee Comment (RC2) · Anonymous Referee #2 · 17 Jul 2019

Shahpoury et al. present a chromatographic method to determine the oxidative potential (OP) of airborne particulate matter. The instrument operates "offline" on PM samples that have been collected, transported back to the lab, and subsequently extracted into a simulated lung lining fluid. Overall, I find the work interesting and likely to be of interest to readers of AMT but substantial revisions and additional work are needed. Specifically, the method was developed and validated with PM standard that is nearly 40 years old; no validation with real-world (contemporary urban PM) was conducted. Further, the introduction contains many speculative statements that are incorrectly presented as facts, which I found especially disconcerting. The manuscript has both grammar and stylistic problems. For example, verb tense is inconsistent through-

out the introduction and several noun objects are singular when they should be plural. I recommend rejection with an invitation to resubmit once the manuscript is improved.

Specific Comments:

1. ORP stands for "oxygen reduction potential" in the broader electrochemistry field and should not be used here as an acronym for "oxidation/reduction potential".

2. Figures 2-6: Kinetic data should be presented on a molar basis (or mass), not a percentage basis because (1) others cannot compare their results directly to yours and (2) some of the method variability may be masked by this approach since outcome measures become normalized to input concentrations.

3. Abstract, line 10: I take issue with this statement. The OP of PM is not a direct measure of inhalation toxicity. It is a measure of the redox activity of the particles in solution. While some researchers have correlated OP with select measures of cellular toxicity, it would be better to say that OP is often used as a "surrogate" to estimate one form of PM toxicity. Lead is a highly toxic metal, but PM made from 100% lead oxide is not likely to have any demonstrative OP associated with it. So if PbO has no OP in solution, does that mean that it's non-toxic??

4. Abstract, line 12: I take issue with this statement. Again, while inflammation of the epithelial tissue is often problematic from a health standpoint, it is not always true that oxidative stress and inflammation are the CAUSE of chronic disease. Inflammation has been associated with chronic diseases (like asthma) but epithelial inflammation is not the cause of all forms of asthma, it is a symptom of an underlying immune disorder.

5. Abstract, line 13: What does "rapid" mean? Be more precise/explicit. Since PM samples must be collected, transported, and extracted, I would not categorize this method as "rapid". The term 'rapid' as used in this context seems a little misleading. The method may be faster than previous reports on methods that examine PM oxidative potential, but the method isn't real-time and also requires physical collection of a PM

sample (which can take several days to complete). Further, the authors never define why this method is "rapid" or how rapid it is....

6. The manuscript defines 15 acronyms in the Introduction section alone. The acronyms themselves are OK but with 15 explicitly defined it becomes nearly impossible to track them down later on. The manuscript needs a list of nomenclature.

7. Intro, page 2, line 3: The term "fine particulate matter" should not be abbreviated as PM but as PM2.5 since the fine refers explicitly to the fine mode.

8. Intro, page 12, line 10: I take issue with the supposition that "inorganic salts" and "crustal dusts" are non-toxic. Crustal dust can contain heavy metals and microorganisms, in addition to elements like crystalline silica - all of which are potentially toxic or inflammogenic. Lead can also exist as an inorganic salt.

9. Intro, page 2, lines 12-14: This statement is outright wrong and the reference is incorrectly used as support. They Ayers article makes no such claim. Indeed, the second to last sentence of the concluding section of the Ayers paper states " If, however, it can be demonstrated that one or more oxidative stress potential tests are better predictors of adverse health outcomes then more traditional mass metrics, there may be a case to base air quality policy upon such tests." While I agree that OP has been shown to correlate well with certain measures of toxicity in vitro, there are only a handful of studies that have demonstrated this behavior in vivo.

10. Intro, page 2, line 14: This statement is incorrect. The OP is more often defined as the ability of the compounds present in solution to act as oxidizing agents.

11. Intro, page 3, line 24. Change "this" to "these". The word data is plural for datum.

12. While the ORP experiments are interesting, they are ancillary to (and largely decoupled from) the main focus of the manuscript. The ORP work should be more thoroughly vetted/explored or removed and submitted as a separate work for publication once properly done.

13. Methods, page 4, line 25: I would hesitate to call SRM 1649b "typical urban PM" today. First, this sample was collected in Washington DC in 1982, so it perhaps represents urban PM from 37 years ago. Second, the NIST sample was a 12-month collection period and was not stored at -80 so all of the semi-volatile material (which likely contributes to PM OP) has likely evaporated. Third, this sample was collected from a large baghouse that was treated with copious amounts of fungicide (run the sample through a GC/MS in scan mode and look for the dichlorophene peak). I recommend additional validation with contemporary PM samples.

14. Figure 2 is not needed. This information can easily be included in one of the results tables or simply in the text alone.

15. Results, page 8, and elsewhere: Several of the findings reported in this work have been reported previously, such as the non-linear effects of PM concentration on thiol oxidation (Charrier and Anastasio, 2016) but those previous reports are not discussed or acknowledged here. A more thorough review (and inclusion) of the recent literature on PM OP is needed.

16. Conclusions, page 10: The authors suggest that AA is a better candidate for determining the OP of ambient PM but fail to recognize that many of the epidemiological studies, to date, have not reported associations between AA-determined OP and various adverse health effects (whereas GSH assays and DTT assays have shown associations with disease). The authors should conduct a more thorough review of the health effects literature before making such a claim.

17. Figure 3 - Why does the ORP of the reference SELF change over time?

18. Figures - Incubation time and PM concentration should be included in the Figure label where relevant.

19. I also think Figure 4B can be deleted since it was shown in Figure 3 that product formation is biased low (from a stoichiometry standpoint) because of likely complex

formation between products and other solution compounds/proteins.

---

## Author Comment (AC1) · 16 Oct 2019

The comment was uploaded in the form of a supplement:
https://www.atmos-meas-tech-discuss.net/amt-2019-220/amt-2019-220-AC1-supplement.pdf

---

## Author Response (AR1)

**Authors' responses to Reviewers' comments**

**Manuscript ID:** amt-2019-220
**Title:** Development of a chromatographic method to study oxidative potential of airborne particulate matter
**Authors:** Pourya Shahpoury, Tom Harner, Gerhard Lammel, Steven Lelieveld, Haijie Tong, Jake Wilson

We would like to thank the reviewers for their valuable comments and suggestions that aimed to improve the quality of our manuscript. We have addressed the reviewers' specific comments below and in the main text (highlighted in yellow).

In addition to the changes made at the Reviewers' request, we have made the following changes:

- New Title: *Development of an antioxidant assay to study oxidative potential of airborne particulate matter*
- Revised author list due to addition of new contents
- Revised abstract and conclusions
- Addition of Supplementary Information
- Additional changes in the text listed at the end of this document

**Reviewer #3 comments**

**1) Reviewer's general comment:**

In this manuscript, Shahpoury et al. report a chromatographic (LC-MS) method of analyzing oxidative potential (OP) of airborne particulate matter (PM) in simulated epithelial lining fluid (SELF). In the introduction section, the authors reviewed the current state of analyzing OP of airborne PM and raised a few existing problems as following.

1. The commonly used DDT method is not a good indicator for OP because its reaction does not well represent those antioxidants (e.g., ascorbic acid, GSH and CSH) in biological fluids.

2. Existing studies using real biological antioxidants to probe OP did not fully consider the auto-oxidation of them and thus carried uncertainty.

3. Existing methods have disadvantages of 1) being time-consuming and expensive, and 2) measurement variability among different assays. Other than these problems, the authors also proposed to explore the validity of using electrochemical potential as an indicator of PM's OP, which I took as the 4th aspect of this paper beside the three problems. Below are my reviewing report with focuses on evaluating how far each aspect (numbered 1-4) above has been addressed.

*Authors' response:*
No corrections required. We have addressed the reviewer's specific comments below:

2) Reviewer's comment:

Aspect 1: a major merit of this study is that the authors evaluated OP with three common antioxidants in SELF and thus made their method more approximating the actual redox environment in biological fluids. LCMS analysis of the three antioxidant and their oxidized products is more accurate in quantification. However, I would recommend the authors to make comparison of their method to DDT method to directly demonstrate the method's advantage.

*Authors' response:*

Following the Reviewer's suggestion, we have now included results describing the dose-response relationship and reproducibility with the DTT assay with SRM1649, which we originally used for the antioxidant assay method development. The DTT method details are included in the Supplement (created during revision process), and the results are described under Section 3.5.

Page 10, Line 11-23:

*"**3.5 Comparison with the dithiothreitol assay**

*In order to evaluate the performance of the current method, we tested the SRM sample with the dithiothreitol assay and concentrations of 25 to 100 µg mL$^{-1}$. The method details and results are presented in the Supplement……………………………………...results from the dithiothreitol assay in the literature (Charrier and Anastasio, 2012; Fang et al., 2016; Crobeddu et al., 2017). Additional work is being conducted to compare the reaction kinetics between the two assays."*

Supplement, Page 4:

*"**Section S1. Method description for the dithiothreitol assay**

*The Dithiothreitol (DTT) assay was performed following the procedure described in Tong et al., (2018). In brief, SRM1649 was incubated in phosphate buffered (PBS)………………………… due to reaction with SRM, is determined indirectly using measured concentrations of TNB$^-$. For data reduction, for each time-step a fresh DTT reference curve was made ranging from 20 to 0 µM DTT."*

3) Reviewer's comment:

Moreover, how is LCMS analysis compared with HPLC with absorbance detection? What is the necessity of using MS instead of a cheaper diode array detector?

*Authors' response:*

We have added the following statement in order to stress the advantage of using UHPLC-MS/MS:

Page 3, Line 29-33:

*"The use of UHPLC-MS/MS allows relatively short analytical time down to few minutes, whereas the MS/MS capability allows simultaneous detection of multiple analytes (see Table 2). This is a major advantage compared to conventional HPLC analysis with absorbance detection, and it is important when performing high-throughput laboratory analysis, in particular when analyzing high-molecular mass substances with various organic or water solubility."*

**4) Reviewer's comment:**

Aspect 2. Page 3, L10-15: Giving a little more detailed explanation on the chemistry of auto-oxidation and how it causes analytical uncertainty will be helpful for readers to understand the issue.

**Authors' response:**

We have added the following statement and provided a reference to address the reviewer's comment:

Page 3, Line 21-24:

"*The artifact can happen during sample processing due to various factors, such as sample acidification or restoration of pH to neutral/alkaline (Rossi et al., 2002), which may lead to variable oxidation of thiols and low reproducibility of measured GSH and GSSG concentrations.*"

**5) Reviewer's comment:**

Page 7, L6-7: Specifying "those reported by Crobeddu et al." is recommended and a quantitative comparison of the "precision" here would make the argument more plausible. I am not fully certain how the auto-oxidation is avoided by the authors. Is it because a "reference" (without PM) is subtracted from a PM-contained sample? By skimming the cited Crobeddu's work, I found that study also considered the subtraction of blank. The authors should better justify the improvement of their method here.

**Authors' response:**

We have added the following statements to the text to address the Reviewer's comment:

Page 3, Line 33-37:

"*Pre-analytical sample preparation is key to achieving optimal results in the analysis of antioxidants. In the case of GSH and CSH, masking the –SH group is crucial in preventing their artificial oxidation prior to instrumental analysis (Giustarini et al., 2016; Escobar et al., 2016). In the present study, this has been achieved by derivatizing GSH and CSH with N-ethylmaleimide (NEM), which is an effective alkylating reagent for masking –SH group at the pH range of 6.5-7.5.*"

Page 7, Line 14-18:

"*The results indicate noticeable enhancement in precision compared to those reported by Crobeddu et al., (2017), where derivatization and blocking of the GSH thiol group was not considered during sample processing and RSD values of >50% were observed in several cases. Re-calculating our results following the Crobeddu et al., (2017) approach (i.e. % depletion), the RSD for GSH depletion with our method would be 8% for inter-sample variability and ≤5% for intra-sample variability.*"

**6) Reviewer's comment:**

Aspect 3. I did not find much demonstration in this paper of the method's advantage in saving analysis time and cost. The use of LCMS will probably increase the cost. It also did not discuss on the analytical variance across different assays.

**Authors' response:**

The aim of our study was to enhance the current oxidative potential methods in terms of analytical capabilities, and performing cost analysis in relation to other methods falls outside the manuscript

objectives. However, in response to the Reviewer's comment No. 3 (please see above), we commented on the advantages of the new method in terms of reduced analytical time. In addition, we have now compared our method performance with the DTT assay in terms of sensitivity and reproducibility, in response to Reviewer's comment No. 2, and to the Crobeddu et al., (2017) antioxidant assay in response to Reviewer's comment No. 5 (please see above).

7) Reviewer's comment:

Aspect 4. I have most concerns on the content related to "electrochemical oxidation-reduction potential (ORP)".

First, on Page 3 L29-30, Flohe's review paper in 2013 was cited, yet the "controversy" was not clarified. In that paper, Flohe argued the point that electrochemical potential does not generate more new information than the concentration ratio of a redox couple, since chemical equilibrium hardly exist in biological environment. I hold the same opinion with Flohe and thus have doubts in the necessity of probing redox potential on top of measuring the concentrations of a redox couple. If the authors can provide their insights into this question, that'll substantiate the use of electrochemical potential beyond simply piling up data.

*Authors' response:*

We have now removed the results and discussion on the experimental 'electrochemical oxidation-reduction potential' following the recommendation by the Reviewer #2. Hence, the suggested changes are not applicable to the revised manuscript.

Under sections 3.3, 3.3.1, and 3.4 (and Figure 4 and 5, and S1b), in addition to the antioxidant depletion rates (or oxidation product formation rates), we have reported the *theoretical redox states* of the simulated lung fluid that we calculated using the measured concentrations of redox couples CSH/CSSC and GSH/GSSG, considering that this is an established method (references provided: Jones et al., 2000; Iyer et al., 2009). We have provided the following justification for this choice:

Page 6, Line 27- Page 7, Line 2:

"*Availability of the concentrations of redox couples CSH/CSSC and GSH/GSSG allows calculating the redox state of SELF before and after addition of PM. In the biochemical literature, the Nernst equation has been frequently used to calculate the reduction potential (Eh, expressed in mV) of antioxidant redox couples ……………………………………………………………………..in addition to presenting the results as molar concentrations, we calculated the theoretical redox state of CSH/CSSC and GSH/GSSG redox couples, Eh (mV), using the molar concentrations (mol $L^{-1}$) of these species (Table S1) measured with the present method and following the Nernst equation (at pH 7.4), Eq. 1 and 2 (Jones et al., 2000; Iyer et al., 2009)*"

8) Reviewer's comment:

Second, more description of the electrode measurement method is required to confirm the validity of the measured ORP. The key missing information includes how long the electrode takes to achieve a stable reading, and what is the principle of determining the stable reading.

*Authors' response:*

As noted above, in the current version, we have removed the contents related to measured electrochemical oxidation-reduction potential (ORP) following the Reviewer #2's suggestion. Hence, the suggested changes are not applicable to the revised manuscript.

In Figure 3, it is obvious that the redox is in non-equilibrium up to 240 min, and I would not expect the time scale for establishing equilibrium between a particle solution and an electrode is much shorter than the redox reaction occurring in the system. In other words, a stabilized OPR is hard to be obtained if redox reactions are taking place. The authors should justify whether the measured OPR is in equilibrium (whose likelihood is low from my perspective).

*Authors' response:*

As noted above, we have removed the contents related to measured electrochemical oxidation-reduction potential (ORP) following recommendation by the Reviewer #2. Hence, the comment above is not applicable to the revised manuscript.

10) Reviewer's comment:

Figure 3c: Which AO species is this figure representing? Is it AA? The decrease of ORP_ref is an indication that the reaction is taking place toward reducing, i.e., the oxidized AO species is gaining electrons. This means oxidized AO species exists at a very initial phase of the reaction. How does this happen? Where is the oxidized AO species from at t0?

*Authors' response:*

As noted above, in the revised manuscript, we have removed the contents related to measured electrochemical oxidation-reduction potential (including Figure 3C) following recommendation by the Reviewer #2. Hence, the above comment is not applicable to the revised manuscript.

11) Reviewer's comment:

Figure 5: What is the solution matrix of this figure? What is the reference of the potential scale? I would recommend the authors provide raw concentration data for calculating Eh.

*Authors' response:*

Figure 5, which contained the measured values of the electrochemical oxidation-reduction potentials (ORP), has been removed during the revision process following recommendation by the Reviewer #2. Regardless, we have now included the solution matrix and the related PM (SRM) concentrations to the figure captions:

"***Figure 2.*** *Depletion of antioxidants, AA, CSH, and GSH, and formation of oxidation products CSSC and GSSG in SELF-a containing 20 $\mu g\ mL^{-1}$ of SRM and following incubation time of 180 min.*"

"***Figure 3.*** *Time-dependent depletion of antioxidants AA, CSH, and GSH, and formation of oxidation products CSSC and GSSG in SELF-a containing 25 $\mu g\ mL^{-1}$ of SRM*"

"***Figure 4.*** *Dose-response of antioxidants AA, CSH, and GSH depletion and oxidation products CSSC and GSSG formation due to increase in SRM concentration in SELF-a (left y-axis) following 180 min incubation*"

"***Figure 5.*** *Application of antioxidant assay to $PM_{2.5}$ samples (A-C) from NAPS station in Hamilton. Samples were incubated in SELF-a for 180 min.*"

"***Figure 6.*** *The effects of SELF composition on antioxidant depletion rates in the presence of 20 $\mu g\ mL^{-1}$ SRM following 180 min incubation. The details of SELF formulations are given in Table 1*"

Moreover, we have now reported the *theoretical redox state* (Eh) of the simulated lung fluid in Figure 4, along with the depletion rates of antioxidants and formation rates of oxidation products in molar unit ($\mu$M min$^{-1}$); we have included the concentration data used for calculating Eh in Table S1 in the Supplement.

12) Reviewer's comment:

Lastly, I have some other supplementary comments as below. In Figure 2, the % depletion and % formation should sum to 100%, if the assumed reaction stoichiometry (Page 6 L 25) is true, but this is not the case. Although the author suggested a possible reason at a later place (Page 7 L30-35), I would recommend it to be discussed and clarified earlier when the stoichiometry firstly appeared.

*Authors' response:*

The statement regarding stoichiometry to which the Reviewer is referring has been omitted following recommendation by Reviewer #2 to report the results in molar unit.

In the revised manuscript, we have discussed the molar consumption rates of CSH and GSH, and formation rates of CSSC and GSSG in the context of stoichiometry when reporting the related data in Sections 3.2 and 3.3.1:

Page 8, Line 12-17:

"*Considering the molar concentrations of the redox pairs CSSC/CSH and GSSG/GSH, the observed formations of CSSC and GSSG were found to be noticeably lower than the theoretical values anticipated from reaction stoichiometry, i.e. 2 CSH → 1 CSSC and 2 GSH → 1 GSSG. This was more pronounced for CSH/CSSC (i.e. 6 → 1) than GSH/GSSG couple (i.e. 3 → 1). Such observations may be related to the formation of complexes between the organic molecules in PM and deprotonated –SH, prior to formation of CSSC or GSSG, such as glutathionylated quinone species (Song and Buettner, 2010).*"

Page 9, Line 19-20:

"*Similarly, CSSC and GSSG formations were lower than anticipated given the reaction stoichiometry, as we also found with the kinetic study (Sect. 3.2), with CSH/CSSC and GSH/GSSG molar ratios of 7.2±1.1 and 3.5±0.1, respectively.*"

13) Reviewer's comment:

Page 3 L 17: What does "the same chemical species" mean? (what chemical species?)

*Authors' response:*

The statement that the Reviewer is referring to has been removed from the revised manuscript following recommendation by the Reviewer #2 to remove the contents related to experimental oxidation-reduction potentials (ORP).

14) Reviewer comment:

Page 7 L 1-2: In the sentence "The mean consumption for AA, ...",is this referring to the Ref sample or PM-contained sample?

*Authors' response:*

We have revised the sentence as follows:

Page 7, Line 9-10:

"*The mean consumption for AA, CSH, and GSH due to the presence of PM were 0.29±0.03, 0.43±0.03, and 0.53±0.04 µM min$^{-1}$, respectively,*"

**Reviewer #2 comments**

Reviewer's general comment:

Shahpoury et al. present a chromatographic method to determine the oxidative potential (OP) of airborne particulate matter. The instrument operates "offline" on PM samples that have been collected, transported back to the lab, and subsequently extracted into a simulated lung lining fluid. Overall, I find the work interesting and likely to be of interest to readers of AMT but substantial revisions and additional work are needed. Specifically, the method was developed and validated with PM standard that is nearly 40 years old; no validation with real world (contemporary urban PM) was conducted. Further, the introduction contains many speculative statements that are incorrectly presented as facts, which I found especially disconcerting. The manuscript has both grammar and stylistic problems. For example, verb tense is inconsistent through- out the introduction and several noun objects are singular when they should be plural. I recommend rejection with an invitation to resubmit once the manuscript is improved.

*Authors' response:*

We have addressed the Reviewer's concerns entirely, including those related to the use of standard reference PM and the quoted statements in the introduction section; we included results from the application of the new assay to contemporary PM2.5 samples, and revised the statements in the introduction following the Reviewer recommendations. Please see below our case-by-case responses to the Reviewer's specific comments.

1) Reviewer's comment:

ORP stands for "oxygen reduction potential" in the broader electrochemistry field and should not be used here as an acronym for "oxidation/reduction potential".

*Authors' response:*

The contents related to the measured oxidation-reduction potential (ORP) have been removed from the manuscript following the recommendation of Reviewer #2. Hence, the suggested change is not applicable anymore.

2) Reviewer's comment:

Figures 2-6: Kinetic data should be presented on a molar basis (or mass), not a percentage basis because (1) others cannot compare their results directly to yours and (2) some of the method variability may be masked by this approach since outcome measures become normalized to input concentrations.

*Authors' response:*

We have revised figures 2, 3, 4, and 6 following the Reviewer's recommendation and are now reporting the results in molar units. The original Figure 5 has been removed based on Reviewer's suggestion to omit contents related to measured oxidation-reduction potential (ORP). Instead, the new Figure 5 presents oxidative potential data related to the application of antioxidant assay with contemporary PM2.5 samples, following the Reviewer's recommendation.

Please see Figure 2-6 (Page 17-19) in the revised manuscript for clarification.

3) Reviewer's comment:

Abstract, line 10: I take issue with this statement. The OP of PM is not a direct measure of inhalation toxicity. It is a measure of the redox activity of the particles in solution. While some researchers have correlated OP with select measures of cellular toxicity, it would be better to say that OP is often used as a "surrogate" to estimate one form of PM toxicity. Lead is a highly toxic metal, but PM made from 100% lead oxide is not likely to have any demonstrative OP associated with it. So if PbO has no OP in solution, does that mean that it's non-toxic??

*Authors' response:*

The related statement has been revised as follows:

Page 1, Line 11-13: "*Oxidative potential is a measure of redox activity of airborne particulate matter (PM) and is often used as a surrogate to estimate one form of PM toxicity. The evaluation of oxidative potential in physiologically relevant environment is always challenging.*"

4) Reviewer's comment:

Abstract, line 12: I take issue with this statement. Again, while inflammation of the epithelial tissue is often problematic from a health standpoint, it is not always true that oxidative stress and inflammation are the CAUSE of chronic disease. Inflammation has been associated with chronic diseases (like asthma) but epithelial inflammation is not the cause of all forms of asthma, it is a symptom of an underlying immune disorder.

*Authors' response:*

During the revision process, we have removed the sentence to which the Reviewer is referring.

5) Reviewer's comment:

Abstract, line 13: What does "rapid" mean? Be more precise/explicit. Since PM samples must be collected, transported, and extracted, I would not categorize this method as "rapid". The term 'rapid' as used in this context seems a little misleading. The method may be faster than previous reports on methods that examine PM oxidative potential, but the method isn't real-time and also requires physical collection of a PM sample (which can take several days to complete). Further, the authors never define why this method is "rapid" or how rapid it is....

*Authors' response:*

The term "rapid" referred to the following word "chromatographic"; with our method, the target analytical data for each sample is obtained in less than 2 minutes, which is much shorter than the previous

chromatographic methods. We have now removed the words "novel rapid" from the sentence in order to avoid confusion.

**6) Reviewer's comment:**

The manuscript defines 15 acronyms in the Introduction section alone. The acronyms themselves are OK but with 15 explicitly defined it becomes nearly impossible to track them down later on. The manuscript needs a list of nomenclature.

*Authors' response:*

We have removed the acronyms from the abstract with the exception of PM. Five acronyms were removed from the Introduction section, including OP, TMs, DTT, AOs, and ORP. Moreover, for clarification, the acronyms SELF, AA, CSH, GSH, UA, and DPPC have now been defined in Table 1 caption, and AA, CSH, GSH, CSSC, GSSG, and NEM have been additionally defined in Table 2 caption. Given these changes, we do not think that providing the list of nomenclature would offer any advantage.

Please see Table 1 and 2 captions on Page 16.

**7) Reviewer's comment:**

Intro, page 2, line 3: The term "fine particulate matter" should not be abbreviated as PM but as PM2.5 since the fine refers explicitly to the fine mode.

*Authors' response:*

The sentence has been revised as follows:

Page 2, Line 3-4: "*Human exposure to fine particulate matter (aerodynamic diameter $\leq$2.5 µm, i.e. $PM_{2.5}$), is one of the main causes of the adverse effects across the world ...*"

**8) Reviewer's comment:**

Intro, page 2, line 10: I take issue with the supposition that "inorganic salts" and" crustal dusts" are non-toxic. Crustal dust can contain heavy metals and microorganisms, in addition to elements like crystalline silica - all of which are potentially toxic or inflammogenic. Lead can also exist as an inorganic salt.

*Authors' response:*

We have revised the sentence as follows:

Page 2, Line 10-12: "*However, a large fraction of PM is made of chemicals with low toxicity, e.g. sea salt, ammonium sulfates and nitrates, and only a small portion of PM including the organic phase and metals are expected to pose toxic effects (Ayres et al., 2008; Lodovici and Bigagli, 2011).*"

**9) Reviewer's comment:**

Intro, page 2, lines 12-14: This statement is outright wrong and the reference is incorrectly used as support. They Ayers article makes no such claim. Indeed, the second to last sentence of the concluding section of the Ayers paper states " If, however, it can be demonstrated that one or more oxidative stress potential tests are better predictors of adverse health outcomes then more traditional mass metrics, there may be a case to base air quality policy upon such tests." While I agree that OP has been shown to correlate well with certain measures of toxicity in vitro, there are only a handful of studies that have demonstrated this behavior in vivo.

*Authors' response:*

The statement to which the Reviewer is referring has been revised as follows:

Page 2, Line 14-18: "*Growing evidence links the adverse health effects of air pollution to pulmonary oxidative stress due to increased exposure to atmospheric oxidants or decreased antioxidant defense levels (Kelly, 2003; Li et al., 2003; Künzli et al., 2006; Borm et al., 2007; Weichenthal et al., 2013; Kelly and Fussell, 2017; Bates et al., 2019). Consequently, the oxidative potential of PM$_{2.5}$ has been applied as an additional parameter to PM mass in explaining the air pollution health effects (Weichenthal et al., 2019).*"

**10) Reviewer's comment:**

Intro, page 2, line 14: This statement is incorrect. The OP is more often defined as the ability of the compounds present in solution to act as oxidizing agents.

*Authors' response:*

We have revised the statement as follows:

Page 2, Line 19-21: "*In the context of inhalation toxicity, oxidative potential is defined as the ability of PM-bound chemicals to oxidize the lung antioxidants and catalytically generate reactive oxygen species, hydrogen peroxide, superoxide radical, and hydroxyl radical (Bates et al., 2019).*"

**11) Reviewer's comment:**

Intro, page 3, line 24. Change "this" to "these". The word data is plural for datum.

*Authors' response:*

We have omitted the sentence to which the Reviewer is referring, following the recommendation to remove the contents related to the measured oxidation-reduction potential (ORP).

**12) Reviewer's comment:**

While the ORP experiments are interesting, they are ancillary to (and largely de-coupled from) the main focus of the manuscript. The ORP work should be more thoroughly vetted/explored or removed and submitted as a separate work for publication once properly done.

*Authors' response:*

We do understand the Reviewer's point. Despite the close links that we have observed in our results, the experiments related to oxidation-reduction potential might be distracting from the focus of the current manuscript. Hence, following the Reviewer's suggestion, we have removed the related contents from the manuscript and are going to present this concept as a separate paper.

**13) Reviewer's comment:**

Methods, page 4, line 25: I would hesitate to call SRM 1649b "typical urban PM "today. First, this sample was collected in Washington DC in 1982, so it perhaps represents urban PM from 37 years ago. Second, the NIST sample was a 12-month collection period and was not stored at -80 so all of the semi-volatile material (which likely contributes to PM OP) has likely evaporated. Third, this sample was collected from a large baghouse that was treated with copious amounts of fungicide (run the sample through a

GC/MS in scan mode and look for the dichlorophene peak). I recommend additional validation with contemporary PM samples.

*Authors' response:*

SRM1649 was used in this work to allow our method and results to be readily reproduced and evaluated by others. We have addressed the Reviewer's concern by providing oxidative potential results with our method using contemporary ambient $PM_{2.5}$ samples collected from Canadian Air Pollution Surveillance Network. The results are now presented (Section 3.3.1 and Figure 5) alongside the original results obtained from SRM1649, and demonstrates consistent performance of the new method:

Page 9, Line 11-23:

"**3.3.1 Method application to PM$_{2.5}$**

*In order to validate the current method with contemporary ambient samples, we applied the method to $PM_{2.5}$ samples collected from the NAPS station in Hamilton (Fig. 5, x-axis A-C). The samples ……………………………………………………………….., with CSH/CSSC and GSH/GSSG molar ratios of 7.2±1.1 and 3.5±0.1, respectively. AA did not follow the CSH and GSH depletion patterns, and consequently the $Eh_\Delta$ (Fig. 5), reaffirming that AA and thiol antioxidants respond to different PM constituents and emission sources (Ayres et al., 2008; Fang et al., 2016; Weichenthal et al., 2019)."*

14) Reviewer's comment:

Figure 2 is not needed. This information can easily be included in one of the results tables or simply in the text alone.

*Authors' response:*

We disagree with this comment; we believe that Figure 2 actually helps visualize the method performance in terms of reproducibility. Instead, we have combined Figure 2A and B into one graph. Please see Figure 2 on page 17.

15) Reviewer's comment:

Results, page 8, and elsewhere: Several of the findings reported in this work have been reported previously, such as the non-linear effects of PM concentration on thiol oxidation (Charrier and Anastasio, 2016) but those previous reports are not discussed or acknowledged here. A more thorough review (and inclusion) of the recent literature on PM OP is needed.

*Authors' response:*

We could not find the specific paper (i.e. Charrier and Anastasio 2016) to which the Reviewer is referring. We suspect that the Reviewer may be referring to *Charrier et al., (2016)*; however, that paper discusses the bias and non-linearity in *mass-normalized* results with the DTT assay, which are not relevant to the discussions made in our manuscript.

Alongside the recent papers that we previously cited (e.g. Bates et al., 2019; Weichenthal et al., 2019; Tong et al., 2018; Calas et al., 2018), we have now added additional relevant contents from the following publications:

Page 2, Line 13: Pöschl and Shiraiwa (2015)

Page 2, Line 15: Kelly (2013); Li et al., (2003)

Page 2, Line 16: Borm et al., (2007); Weichenthal et al., (2013)

Page 2, Line 28: Tong et al., (2016; 2017; 2019)

Page 2, Line 37: Fang et al., (2019)

Page 3, Line 23: Rossi et al., (2002)

**16) Reviewer's comment:**

Conclusions, page 10: The authors suggest that AA is a better candidate for determining the OP of ambient PM but fail to recognize that many of the epidemiological studies, to date, have not reported associations between AA-determined OP and various adverse health effects (whereas GSH assays and DTT assays have shown associations with disease). The authors should conduct a more thorough review of the health effects literature before making such a claim.

**Authors' response:**

The statement mentioned by the Reviewer referred to suitability of ascorbic acid in terms of covering a wider PM concentration range only (and not its link to air pollution health outcomes). We understand that the statement in question could be misunderstood; hence, we have removed it from the conclusions.

**17) Reviewer's comment:**

Figure 3 - Why does the ORP of the reference SELF change over time?

**Authors' response:**

Following the Reviewer's recommendation, we have removed the contents related to the oxidation-reduction potential (ORP) measurement, including Figure 3C, to which the Reviewer is referring. Hence, the comment does not apply to the revised manuscript.

**18) Reviewer's comment:**

Figures - Incubation time and PM concentration should be included in the Figure label where relevant.

**Authors' response:**

We have added the PM concentrations and incubation times to figure captions, where relevant. Please see Figures 2-6 on page 17-19:

"*Figure 2. Depletion of antioxidants, AA, CSH, and GSH, and formation of oxidation products CSSC and GSSG in SELF-a containing 20 μg mL$^{-1}$ of SRM and following incubation time of 180 min.*"

"*Figure 3. Time-dependent depletion of antioxidants AA, CSH, and GSH, and formation of oxidation products CSSC and GSSG in SELF-a containing 25 μg mL$^{-1}$ of SRM*"

"*Figure 4. Dose-response of antioxidants AA, CSH, and GSH depletion and oxidation products CSSC and GSSG formation due to increase in SRM concentration in SELF-a (left y-axis) following 180 min incubation*"

"*Figure 5. Application of antioxidant assay to PM$_{2.5}$ samples (A-C) from NAPS station in Hamilton. Samples were incubated in SELF-a for 180 min.*"

"***Figure 6.*** *The effects of SELF composition on antioxidant depletion rates in the presence of 20 µg mL$^{-1}$ SRM following 180 min incubation. The details of SELF formulations are given in Table 1*"

19) Reviewer's comment:

I also think Figure 4B can be deleted since it was shown in Figure 3 that product formation is biased low (from a stoichiometry standpoint) because of likely complex formation between products and other solution compounds/proteins.

*Authors' response:*

In the revised manuscript, we have combined Figure 4A and B into one graph (now Figure 4). Regardless of the observation of a low oxidation product formation, we think that it is important to present the molar concentrations of CSSC and GSSG in Figure 4, because Figure 4 presents the dose-response data, whereas Figure 3 shows the reaction kinetics.

**Additional changes to the text:**

Page 2, Line 25-29:

"*Although quinones and transition metals have been recognized as important redox-active substances, other redox-active chemicals, such as humic-like substances, organic hydroperoxides, highly oxygenated molecules, and environmentally persistent free radicals were also found to have oxidative potential (Charrier and Anastasio, 2012; Verma et al., 2015; Dou et al., 2015; Tong et al., 2016; 2017; 2019; Gonzalez et al., 2017).*"

Page 4, Line 1-2:

"*We compared the current method with the dithiothreitol assay and applied the method to contemporary PM2.5 samples.*"

Page 4, Line 30 – Page 5, Line 2:

"*In addition, PM$_{2.5}$ samples were obtained from the National Air Pollution Surveillance (NAPS) site in Hamilton, Canada and used for validating the antioxidant assay. The samples …………………………… concentrations for these samples ranged between 14.3 and 16.1 µg m$^{-3}$. Lab blanks consisted of quartz fiber filters that were not used for air sampling.*"

Page 5, Line 18-19:

"*For experiments involving PM$_{2.5}$ samples from the NAPS site, a 1-cm$^2$ filter punch was added to SELF.*"

Page 8, Line 24-26:

"*The CSH and GSH depletion rates were linear in the PM concentration of 10-40 µg mL$^{-1}$, and slowed down beyond this point; the two analytes were found completely consumed at PM concentration of ≥60 µg mL$^{-1}$ (Fig. 4).*"

Page 9, Line 2-3:

"*This plateau is explained by the complete loss of CSH and GSH at PM concentration of ≥60 µg mL$^{-1}$.*"

Page 9, Line 5-10:

"*We estimated the theoretical redox state of CSH/CSSC and GSH/GSSG redox pairs (Eh) using the Nernst equation (Eq. 1 and 2 in Sect. 2.4; Jones et al., 2000; Iyer et al., 2009). The results are………………………… demonstrates that the oxidative potential of ambient PM can be presented as Eh, which may be more meaningful because Eh represents a redox couple as opposed to a single antioxidant.*"

Page 10, Line 4-10:

"*The addition of DPPC, however, led to slightly better reproducibility of antioxidant depletion rates (Fig. 6), which is consistent with previous reports (Calas et al., 2017). Overall, our results…………………………..a SELF that better represents the true human ELF should be used, rather than the simplified formulas that have been used in the past.*"

Additional revision related to the change of reporting unit from % to molar concentrations (highlighted in yellow in the text)

Page 8, Line 5, 8, 9, 11, 12, 29

Page 9, Line 27, 29, 30, 34

Page 10, Line 3-4

---

## Referee Report (RR1)

The authors have addressed my comments to proper degrees and I recommend the manuscript's publication as is.